# Automated curricula through setter-solver interactions

**Sébastien Racanière, Andrew K. Lampinen**[*]
Equal Contributions
DeepMind
`sracaniere@google.com, lampinen@stanford.edu`

**Adam Santoro, David P. Reichert, Vlad Firoiu, Timothy P. Lillicrap**
DeepMind
`{adamsantoro,reichert,vladfi,countzero}@google.com`

## ABSTRACT

Reinforcement learning algorithms use correlations between policies and rewards to improve agent performance. But in dynamic or sparsely rewarding environments these correlations are often too small, or rewarding events are too infrequent to make learning feasible. Human education instead relies on curricula–the breakdown of tasks into simpler, static challenges with dense rewards–to build up to complex behaviors. While curricula are also useful for artificial agents, handcrafting them is time consuming. This has lead researchers to explore automatic curriculum generation. Here we explore automatic curriculum generation in rich, dynamic environments. Using a setter-solver paradigm we show the importance of considering goal validity, goal feasibility, and goal coverage to construct useful curricula. We demonstrate the success of our approach in rich but sparsely rewarding 2D and 3D environments, where an agent is tasked to achieve a single goal selected from a set of possible goals that varies between episodes, and identify challenges for future work. Finally, we demonstrate the value of a novel technique that guides agents towards a desired goal distribution. Altogether, these results represent a substantial step towards applying automatic task curricula to learn complex, otherwise unlearnable goals, and to our knowledge are the first to demonstrate automated curriculum generation for goal-conditioned agents in environments where the possible goals vary between episodes.

## 1 INTRODUCTION

Reinforcement learning (RL) algorithms use correlations between policies and environmental rewards to reinforce and improve agent performance. But such correlation-based learning may struggle in dynamic environments with constantly changing settings or goals, because policies that correlate with rewards in one episode may fail to correlate with rewards in a subsequent episode. Correlation-based learning may also struggle in sparsely rewarding environments since by definition there are fewer rewards, and hence fewer instances when policy-reward correlations can be measured and learned from. In the most problematic tasks, agents may fail to begin learning at all.

While RL has been used to achieve expert-level performance in some sparsely rewarding games (Silver et al., 2016; OpenAI, 2018; Vinyals et al., 2019), success has often required carefully engineered curricula to bootstrap learning, such as learning from millions of expert games or hand-crafted shaping rewards. In some cases self-play between agents as they improve can serve as a powerful automatic curriculum for achieving expert or superhuman performance (Silver et al., 2018; Vinyals et al., 2019). But self-play is only possible in symmetric two-player games. Otherwise humans must hand-design a curriculum for the agents, which requires domain knowledge and is time-consuming, especially as tasks and environments grow in complexity. It would be preferable to have an algorithm that could automatically generate a curriculum for agents as they learn.

---

[*]DeepMind and Stanford University Department of Psychology

Several automatic-curriculum generating algorithms have been proposed, including some that help agents explore and learn about their environments (e.g. Gregor et al., 2016b; Eysenbach et al., 2018), and some that attempt to gradually increase goal difficulty (e.g. Florensa et al., 2017). Most of these approaches have been tested only on simple tasks in simple environments, and often assume that either the environment is fixed from one episode to the next or that the agent's goal is fixed and unchanging. Ideally, curricula would apply to complex, varying environments and would support goal-conditioning to handle changing tasks.

Surprise- or difficulty-based exploration may sometimes discover desired agent behaviors (Gregor et al., 2016b; Burda et al., 2018; Haber et al., 2018). This approach may not always be practical, though, since many difficult, but otherwise irrelevant tasks might "distract" exploration objectives. For example, training a self-driving car to successfully do flips might be challenging and novel, but it would not be particularly beneficial. Human curricula efficiently lead learners towards a desired competency, rather than along arbitrary dimensions of difficulty. Analogously, it would be useful for algorithms to leverage knowledge of the desired goal distribution to develop more targeted curricula.

This paper take several steps toward automatic, targeted curriculum generation by proposing an algorithm for training a goal-conditioned agent in dynamic task settings with sparse rewards. The approach trains a "setter" model to generate goals for a "solver" agent by optimizing three setter objectives: discovering the subset of expressible goals that are valid, or achievable by an expert solver (*goal validity*), encouraging exploration in the space of goals (*goal coverage*), and maintaining goal feasibility given the agent's current skill (*goal feasibility*). We also propose an extension for targeting a distribution of desired tasks (if one is known) using a Wasserstein discriminator (Arjovsky et al., 2017). We demonstrate our approach in a rich 3D environment and a grid-world wherein observation statistics and possible goals vary between episodes, and show that it substantially outperforms baselines, lesions and prior approaches.

## 2 RELATED WORK

**Uniform sampling of sub-tasks** Perhaps the simplest curriculum is training uniformly over sub-tasks of varying difficulty. For example, Agostinelli et al. (2019) trained a Rubik's cube solver on problems sampled uniformly between 1 and $K$ moves from the solved state. This curriculum leverages the fact that some sub-tasks can be solved before others, and that learning of these sub-tasks bootstraps learning of harder sub-tasks, and ultimately the task as a whole. However, in complex settings uniform training may not suffice, either because easier sub-tasks do not exist, they are still too hard to learn, or they do not help learning of harder sub-tasks. When uniform sampling is ineffective, hand-engineered curricula may work (Elman, 1993; Bengio et al., 2009; Zaremba & Sutskever, 2014; Graves et al., 2016). Their effectiveness has led to research on *automated* ways to derive curricula (Graves et al., 2017). Here we outline a number of such approaches in the RL setting.

**Exploration** Some work leverages exploration to encourage state diversity (Gregor et al., 2016b), state-transition surprise (Burda et al., 2018; Haber et al., 2018), or distinguishable skills (Eysenbach et al., 2018). These exploration-based methods are usually validated in relatively simple, unchanging environments, and have not been tested as pre-training for goal-conditioned RL tasks. A few studies have considered varying environments; e.g. Wang et al. (2019) considered evolving environments together with paired agents. However, because each agent is paired to a single environment, the method results in agents that are specialized to single, unchanging environments with fixed goals.

**Optimal task selection** Other approaches include selecting tasks on which learning is progressing (or regressing) the fastest (Baranes & Oudeyer, 2013). However, it can be prohibitively expensive to determine goal regions and track progress within them, especially as task spaces grow larger and more complex. Some approaches work for a set of pre-specified tasks (Narvekar & Stone, 2019), but they require human effort to hand-select tasks from this set. Again, these approaches have also generally been demonstrated in simple, fixed environments.

**Agent-agent interactions** Agent interactions can also generate effective curricula. For example, in symmetric two-player (or two-team) zero-sum games agents jointly improve and thus are forced to face stronger and stronger opponents. This natural curriculum may work on tasks where random play can achieve rewards with reasonable frequency (Silver et al., 2018). But in other cases, hand-engineered auxiliary tasks may be used to avoid the difficult initial problem of learning from sparse

rewards, such as imitation learning on data from experts (Silver et al., 2016; Vinyals et al., 2019). Or, dense shaping rewards may be needed (OpenAI, 2018; Jaderberg et al., 2019). Furthermore, this type of curriculum has not been tested in goal-conditioned environments – while the environment might vary because of opponent play, or on a different map, the ultimate goal of winning is fixed. More fundamentally, while this type of curriculum works well for two-player zero-sum games, it is less clear how it can be used to train a single agent on a non-competitive, goal-conditioned task.

Asymmetric agent-agent interactions, for example when one agent tries to repeat or undo another's actions (Sukhbaatar et al., 2017), can also be useful. However, this requires the desired task distribution to be close to the distribution generated by these reversing/repeating tasks. In goal-conditioned settings, guaranteeing this is likely as difficult as the original learning problem.

**Goal conditioning**  In the goal-conditioned setting, hindsight experience replay (Andrychowicz et al., 2017) has agents retrospectively imagine that they were trying to achieve the state they actually ended up in. While this is an active curriculum for starting learning, it does not necessarily encourage goal-space exploration, nor does it provide a framework for generating novel goals.

Nair et al. (2018) used a generative model of state space to sample "imagined" goals, rewarding the agent based on similarity to the generative model's latent space. Florensa et al. (2017) used a GAN to generate goals of intermediate difficulty for the agent, which resulted in goals that gradually expanded to fill the goal space. This work is closely related to part of our proposal, and we use it as an important benchmark. Critically, this approach has not been tested in environments which vary substantially from episode to episode, particularly ones where the valid goals change from episode to episode. This is an important distinction because training generative models with non-trivial conditioning can be challenging. In particular, while conditioning directly on an informative latent variable can work well, for example when trying to generate images from a given class (Mirza & Osindero, 2014; Brock et al., 2018), even this problem is not completely solved (Ravuri & Vinyals, 2019). Adding the challenge of trying to *discover* latent variables with which to condition and performing even a simple manipulation of them makes things much more difficult (Rezende & Viola, 2018) (c.f. the difficulty of learning *hierarchies* of latent variables (Sønderby et al., 2016; Maaløe et al., 2019)). This means that if the valid goals are not trivially observable from the environment, it may be difficult for the goal-setter to discover the goal structure via a generative loss alone. In section 4.2, we demonstrate this particular failure mode, along with some successes.

**Summary**  A variety of automated curriculum generation approaches for RL have demonstrated some success, but the challenge of curriculum generation in the more complex settings remains open. This is because these approaches have not demonstrated success in tasks with the complexity reflective of difficult real-world tasks; in particular, no approach can handle goal-conditioned tasks in dynamic environments, wherein the set of possible goals varies from one episode to the next, and the set of possible goals might be tiny compared to the set of expressible goals.

## 3 METHOD

Our model consists of three main components: A **solver** – the goal-conditioned agent we are training. A **setter** ($S$) – A generative model we are using to generate a curriculum of goals for the agent. A **judge** ($J$) – A discriminative model that predicts the feasibility of a goal for the agent at present. See appendix B for architectural details.

See fig. 1 for training schematics (see also Appendix B.2). The solver agent trains on setter-generated goals using a distributed learning setup to compute policy gradients (Espeholt et al., 2018). For setter training, three concepts are important: goal *validity*, goal *feasibility* and goal *coverage*. We say a goal is *valid* if there exists a solver agent policy which has a non-zero probability of achieving this goal. This concept is independent of the current policy of the solver. By contrast, *feasibility* captures whether the goal is achievable by the solver at present. Specifically, we say a goal has feasibility $f \in [0, 1]$ if the probability that the solver will achieve the goal is $f$. The set of feasible goals will therefore evolve as the solver learns. The judge is a learned model of feasibility, trained via supervised learning on the solver's results. Finally, goal *coverage* indicates the variability (entropy) of the goals generated by the setter.

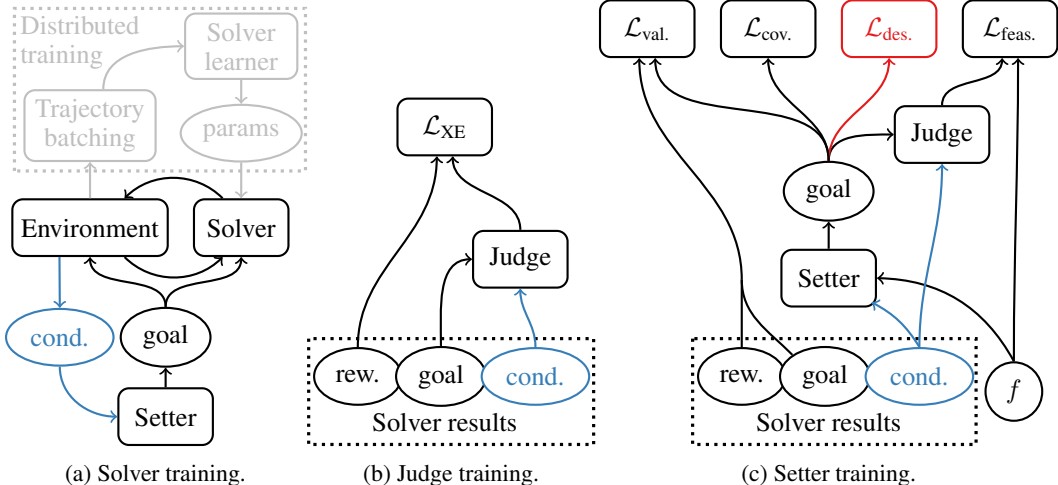

Figure 1: Training schematic (see also appendix B.2). The setter and judge only receive conditioning observations of the environment (blue) when the environment varies across episodes. The desirability loss (red) can be used when the distribution of desired tasks is known *a priori*.

### 3.1 REWARD AND LOSSES FOR THE SOLVER

Our solver is a goal conditioned RL agent. At the beginning of every episode it receives a goal $g$ sampled by the setter, and a single reward $R_g$ at the end of the episode. The reward $R_g$ is 1 if the solver achieved the goal, or 0 if it did not after a fixed maximum amount of time. The solver could be trained by any RL algorithm. We chose to adopt the training setup and losses from Espeholt et al. (2018). The solver consists of a policy $\pi$ and a baseline function $V^\pi$ which are trained using the V-trace policy gradient actor-critic algorithm with an entropy regularizer (see Espeholt et al. (2018) for details).

### 3.2 LOSS FOR THE JUDGE

The Judge $J$ is trained as a binary classifier to predict the reward, 0 or 1. Given a goal $g$ (see section 3.3), $J(g)$ are logits such that $\sigma(J(g)) = p(R_g = 1|g)$, where $\sigma$ is the sigmoid function, $R_g$ are returns obtained by the solver when trying to achieve those goals, and $p(R_g = 1|g)$ is the probability assigned by the judge that the agent will have a return of 1 when given goal $g$. We use a cross-entropy loss with the input distribution defined by the setter, and labels are obtained by testing the solver on these goals:

$$\mathcal{L}_{Judge} = -\mathbb{E}_{z \in \mathcal{N}(0,1), f \in \text{Unif}(0,1), g = S(z,f)} \left[ R_g \log(\sigma(J(g)) + (1 - R_g)(\log(1 - \sigma(J(g)))) \right]$$

### 3.3 LOSSES FOR THE SETTER

Our setter takes as input a desired goal feasibility $f \in (0, 1)$. In particular, we can sample a goal $g = S(z, f)$ for some sample $z$ from a Gaussian prior $\mathcal{N}(0, 1)$ and a desired feasibility $f$, or we can map backwards from a goal $g$ to a latent $z = S^{-1}(g, f)$, for which we can then compute the probability under the prior. Both directions are used in training. With these features in mind, we define three losses for the setter that reflect the concepts of goal validity, feasibility, and coverage:

**Validity:** A generative loss that increases the likelihood of the setter generating goals which the solver has achieved. This is analogous to the hindsight of Andrychowicz et al. (2017), but from the setter perspective rather than the solver. Specifically:

$$\mathcal{L}_{\text{val.}} = \mathbb{E}_{g \text{ achieved by solver}, \xi \in \text{Unif}(0,\delta), f \in \text{Unif}(0,1)} \left[ -\log p \left( S^{-1}(g + \xi, f) \right) \right]$$

Where $g$ is sampled from goals that the solver achieved, regardless of what it was tasked with on that episode, $\xi$ is a small amount of noise to avoid overfitting[1], and $p(.)$ denotes the probability of

---

[1]This is common practice in generative models of images or discrete data. See e.g. Gregor et al. (2016a)

sampling that latent under a fixed gaussian prior for the latent of $S$. This loss may not cover all valid goals, but it is a good estimate available without any other source of knowledge.

**Feasibility:** A loss that encourages the setter to choose goals which match the judge's feasibility estimates for the solver at present. Specifically:

$$\mathcal{L}_{\text{feas.}} = \mathbb{E}_{z \in \mathcal{N}(0,1), f \in \text{Unif}(0,1)} \left[ (J(S(z,f)) - \sigma^{-1}(f))^2 \right]$$

This loss uniformly samples a desired feasibility $f$ (to train the setter to provide goals at a range of difficulties), then attempts to make the setter produce goals that the judge rates as matching that desired feasibility. Note although gradients pass through the judge, its parameters are not updated.

**Coverage:** A loss that encourages the setter to pick more diverse goals. This helps the setter to cover the space of possible goals, and to avoid collapse. Specifically:

$$\mathcal{L}_{\text{cov.}} = \mathbb{E}_{z \in \mathcal{N}(0,1), f \in \text{Unif}(0,1)} \left[ \log p \left( S(z,f) \right) \right]$$

This loss maximises the average of the conditional entropy of the setter. Since the density of $f$ is constant, adding a term $\log(p(f))$ in the above formula only changes the loss by a constant, and shows that our loss is equivalent to maximising the entropy of the joint distribution $(S(z,f), f)$.

The setter is trained to minimize the total loss $\mathcal{L}_{\text{setter}} = \mathcal{L}_{\text{val.}} + \mathcal{L}_{\text{feas.}} + \mathcal{L}_{\text{cov.}}$. Note that the sum $\mathcal{L}_{\text{feas.}} + \mathcal{L}_{\text{cov.}}$ can be interpreted as a KL-divergence between an energy model and the setter's distribution. Specifically, for a fixed feasibility $f$, define an energy function on the space of goals by $E_f(g) = (J(g) - \sigma^{-1}(f))^2$. Let $p_f(g) = e^{-E_f(g)}/Z$ be the density of the distribution defined by this energy, where $Z$ is a normalizing constant. Then the sum of the feasibility and coverage losses is, up to a constant, the average over $f \in [0,1]$ of the divergence $KL(p_f || p(S(g,f)))$.

We also demonstrate two important extensions to our framework which are critical in more complicated environments:

**Variable environments and conditioned setters:** While prior work has often focused on fixed environments, such as the same maze each episode, we would like to train agents in variable worlds where the possible goals vary from one episode to the next. For this to be possible, our setter and judge must condition on an environmental observation. However, learning these conditional generative models can be challenging if the valid goals are not trivially observable (see the related work section above). We demonstrate the success of our approach in these environments, and advantages with a conditioned setter and judge.

**Desired goal distributions:** In complex task spaces, the goals we want agents to accomplish will likely lie in a small region within the space of all possible goals. Thus it may not be efficient to uniformly expand difficulty. We propose an additional loss for optimizing the setter towards a desired goal distribution, when such a distribution is known. Specifically, we propose training a Wasserstein discriminator (Arjovsky et al., 2017) to discriminate setter-generated goals from goals sampled from the desired goal distribution. The Wasserstein discriminator has the beneficial property that it can give useful gradients even when the distributions are non-overlapping, which is critical in this setting, since the easy goals the setter generates initially may not have any overlap with the target goal distribution. Specifically, the desirability discriminator loss is:

$$\mathcal{L}_{\text{disc.}} = \mathbb{E}_{g \in \text{desired goal distribution}} \left[ D(g) \right] - \mathbb{E}_{z \in \mathcal{N}(0,1), f \in \text{Unif}(0,1)} \left[ D(S(z,f)) \right]$$

and the setter is trained with the loss:

$$\mathcal{L}_{\text{des.}} = \beta_{\text{des.}} \mathbb{E}_{z \in \mathcal{N}(0,1), f \in \text{Unif}(0,1)} \left[ D(S(z,f)) \right]$$

Where $\beta_{\text{des.}}$ is a hyperparameter. While targeting the desired distribution can be helpful, it is usually not sufficient on its own – the desired tasks may be infeasible at first, so the other setter losses are needed to develop a feasible curriculum. The desirability loss just tries to aim this curriculum in the right direction.

## 3.4 Environments

We work in two environments, which are briefly described below (see appendix C for further details). In each, the solver receives a goal as input during each episode, which it must attempt to achieve.

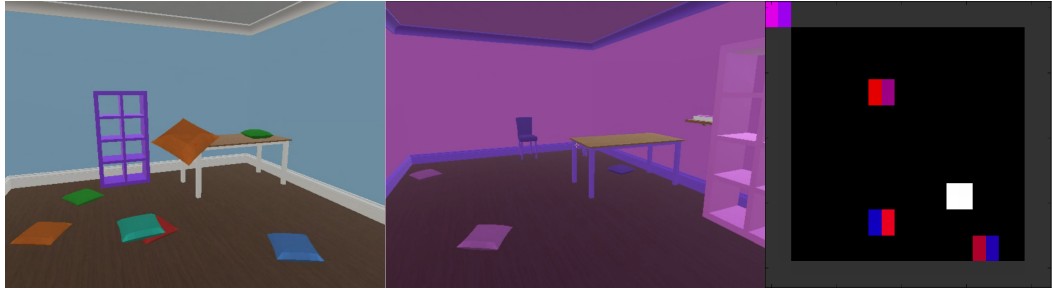

| (a) 3D color-finding. | (b) Recolored color-finding | (c) Grid-world alchemy. |

Figure 2: Our environments. For the 2D grid-world task, the solver (white square) can pick-up objects (bi-colored squares). The object it is currently carrying is displayed in the upper left.

**3D color finding:** A semi-realistic 3D environment built in Unity (http://unity3d.com), consisting of a room containing colored objects and furniture (fig. 2a). The agent can move and look around, and can pick up, manipulate, and drop objects. This results in a complex 46-dimensional action space. Objects and furniture are randomly placed around the room at the beginning of each episode. The agent receives a color (or pair of colors) as a goal, and is rewarded if a patch (or two adjacent patches) in the center of its view contain average colors close to this goal. Both of these tasks sometimes require complex behavior[2]. For example, the agent might have to pick up an object of a yellow color, move it to an object of a blue color and look in between to obtain a green that was not otherwise present in the room. Our agents trained within our framework do indeed exhibit these behaviors. For our extensions, we also used a version of this environment in which the walls, ceiling, and floor of the room, as well as all objects, are procedurally recolored into one of two randomly chosen colors each episode (fig. 2b). This makes the achievable colors in each episode lie in a small subset of color space that overlaps little, if at all, with the achievable colors in other episodes.

**Grid-world alchemy:** A 2D grid world environment, containing a variety of two-colored objects (fig. 2c). The colors of the objects are randomly sampled each episode. The solver can move around the grid, and can walk over an object to pick it up. It cannot put down an object once it has picked it up. If it is already carrying another object, the two objects will systematically combine to make a new object (specifically, the colors are combined by a component-wise max). The solver receives a goal object as input, and is rewarded if it produces a similar object. Because of the combinatorics of the possible object combinations, the irreversibility of picking an object up, and the difficulty of inferring the result of combining two objects, this environment is challenging for both the setter and the solver. Both have the challenging task of learning what is achievable in any particular episode, since each episode contains colors never seen before.

**Evaluation:** In each experiment we evaluate on a fixed test distribution of tasks, regardless of what setter is used for training, in order to have a fair comparison between conditions. In both environments, the space of valid tasks (that could be done by an expert) occupies a small volume in the space of tasks expressible by the setter. In the colour-finding tasks, we do not even know which goals are valid, because of color averaging, shadows, etc. We therefore test on the full set of expressible goals (most of which are invalid), but report performance as a % of best observed scores.

## 4    EXPERIMENTS[3]

### 4.1    COMPLEX ENVIRONMENTS REQUIRE ALL THREE LOSSES

---

[2]See video in https://drive.google.com/drive/folders/1ue8EnmPTQyN9aBlUocw2ZPtVvyxNB-QS?usp=sharing.

[3]To help with reproducibility, we provide code for the networks used for the Setter: https://drive.google.com/drive/folders/1yjhztFeX67tHEImXCiP_UAQfQ-wFvV4Y?usp=sharing.

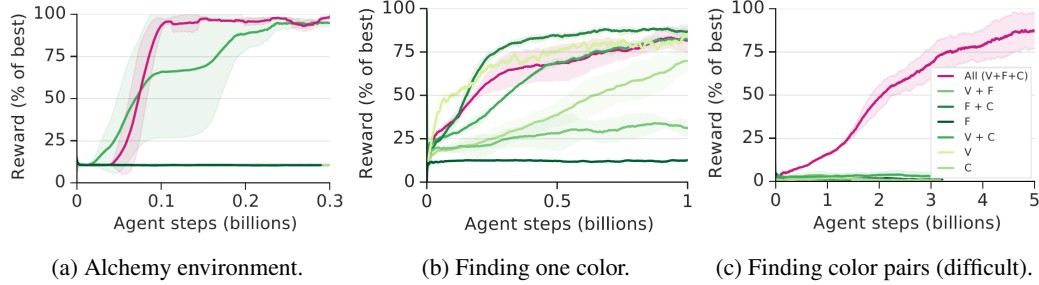

(a) Alchemy environment.  (b) Finding one color.  (c) Finding color pairs (difficult).

Figure 3: All three of our setter losses (V: validity, F: feasibility, C: coverage) are necessary (pink) in complex environments, but various subsets (green) suffice in simpler environments. (Curves plotted are rewards on random evaluation tasks averaged across three runs per condition ± standard deviation across runs; performance is averaged over the last 1000 trials for (a-b), and 5000 for (c). We plot performance as a % of the best agent performance.)

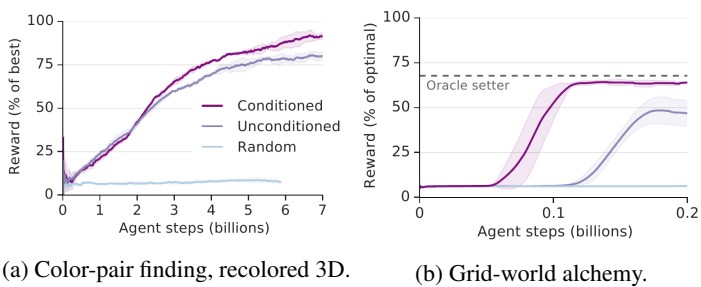

(a) Color-pair finding, recolored 3D.  (b) Grid-world alchemy.

Figure 4: In varying environments, setters that condition on environment observations outperform unconditioned setters, but unconditioned setters are still much better than random goals. (Curves plotted are evaluation rewards averaged across three runs per condition ± standard deviation across runs; performance is averaged over the last 15000 trials for (a), and 1000 for (b).)

## 4.1 Complex environments require all three losses

First, we demonstrate that it is necessary to consider all of goal validity, feasibility, and coverage in complex environments (fig. 3). In the alchemy environment the validity and coverage losses are necessary, while the feasibility loss is not necessary, but does improve consistency (fig. 3a). In the 3D single-color-finding task, various subsets of the losses suffice for learning the task (fig. 3b). However, when the agent must find color pairs, fewer goals are possible and achieving a goal more often requires difficult manipulation of objects. Removing any of the setter losses results in substantially worse performance (fig. 3c). See Appendix B.3 for further analysis of the losses, and supplemental fig. 9 for a visualization of the generated curriculum on a simple location-finding task.

## 4.2 Environments that vary require observation conditioning

### 4.2 Environments that vary require observation conditioning

While much prior work in automated curriculum generation focused on varying goals within a fixed environment, we would like RL systems to perform well on varied tasks in varied environments. For this, they will need to experience a diversity of environments during training, creating the unique challenge of generating curricula that take into account both the current environment and the current abilities of the agent.

To address this we implemented a randomly colored version of our color-finding environment, and the grid-world alchemy task. In both, the set of possible goals changes each episode. We compare a version of our algorithm in which the setter and judge condition on an environmental observation before generating (or evaluating) a task to the basic unconditioned version used in the previous experiments, as well as a random baseline (fig. 4). Solvers trained by the basic version of our

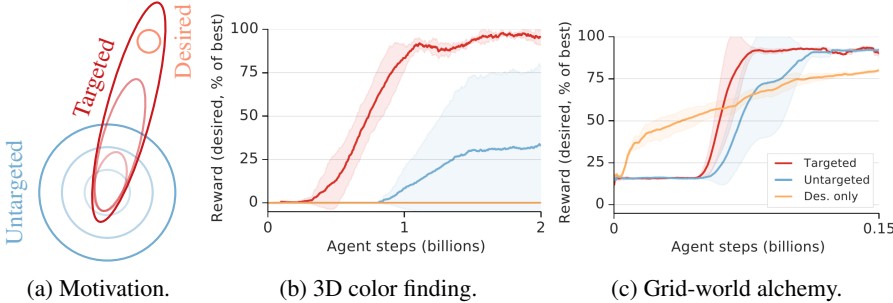

(a) Motivation.    (b) 3D color finding.    (c) Grid-world alchemy.

Figure 5: Targeting a desired goal distribution, when one is known. (Averages across three runs per condition ± std. dev. across runs. Performance is averaged over the last 5000 and 1000 trials for (b) and (c), respectively.)

model still outperform those trained with randomly generated goals. However, the version of our model which conditions on an observation results in better solver performance. To the best of our knowledge, these are the first results demonstrating the success of any automated curriculum approach for goal-conditioned RL in a varying environment.

There are a few points worth noting about our results in the alchemy environment. First, the unconditioned setter had a tendency to not produce stable solver performance. Solver performance would generally degrade after reaching a maximum, while the conditioned setter was able to more steadily maintain solver performance. This was observed across a variety of hyperparameter settings, and merits further investigation. Second, even our conditioned setters are not leading the agents to perfect performance on this task.

However, in grid-world alchemy, the conditioned setter teaches the solver to reach performance close to that of a solver trained by an oracle which samples from the true distribution of possible tasks (fig. 4b). This suggests the limitation is not our setter algorithm, but rather the limitations of the solver agent, for example, the fact that it lacks features like planning (Racanière et al., 2017) or relational inductive biases (Zambaldi et al., 2019) that have proven useful for similar tasks.

In more complex settings the setter may also need auxiliary supervision or stronger inductive biases to overcome the challenges of learning conditional generative models. Indeed, we found that conditioning on a compressed representation (closer to the latent variables) in the recolored color-finding environment gave better results than conditioning on raw observations (see Fig. 10 in the Appendix). Furthermore, in more complex versions of the alchemy environment (for example, introducing more objects with more colors), even our conditioned setter algorithm could not learn to reliably generate feasible goals from raw observations. These results again highlight the challenges of learning conditional generative models when conditioning requires extracting latent variables and performing complex relational reasoning. This will be an important area for future work. Despite this caveat, the success of our setter-solver approach in varied environments represents an important step towards generating curricula in environments closer to the richness of the real world.

### 4.3 TARGETING A DESIRED GOAL DISTRIBUTION IS MORE EFFICIENT

### 4.3 Targeting a desired goal distribution is more efficient

In complex task environments discovering desired behaviors through difficulty-based exploration may not be feasible. There may be many ways a task can be difficult, most of which are irrelevant to what we would ultimately like the agent to achieve. By targeting the desired goal distribution with our desired-goal loss, the setter can push the solver toward mastering the desired tasks more efficiently (fig. 5a). In reality, the path will not be perfectly direct, as the setter trades off feasibility, possibility, and coverage with targeting the desired tasks. However, it will generally be more efficient than untargeted setting, or training on only the desired tasks (if they are difficult).

We first explore this in the 3D color-finding environment. We target a distribution of pairs of 12 bright colors. These pairs are rarely achieved by a random policy, so discovering them is difficult without a setter. Training on only the desired distribution thus results in no learning. The untargeted

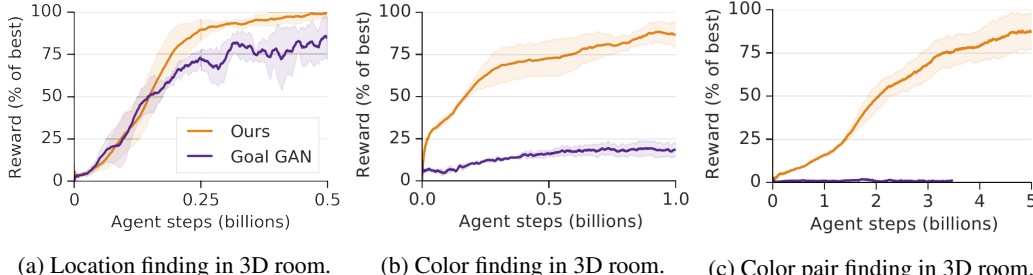

(a) Location finding in 3D room.     (b) Color finding in 3D room.     (c) Color pair finding in 3D room.

Figure 6: Comparison to the Goal GAN (Florensa et al., 2017), the closest approach to our own. (Curves plotted are averages across three runs per condition $\pm$ standard deviation across runs; performance is averaged over the last 1000 trials at each step for (b), and the last 5000 for (a,c).)

setter-solver setup does eventually learn these tasks. However, with targeting it discovers them much more rapidly (fig. 5b), and has a lasting advantage over the untargeted version (see supp. fig. 7).

In the alchemy environment, the story is somewhat different (fig. 5c). We chose the desired distribution to be the most difficult tasks in the environment, consisting of combining half the objects in the room. However, because the setter has the difficult challenge of learning the conditional generative distribution (which is built in to the desired distribution), we find that learning from the desired distribution (if available) results in earlier learning. This is in contrast to the 3D color finding environment, where the desired distribution alone resulted in no learning. This again highlights the complexity of learning to generate goals when the valid goal distribution is conditional in complex, non-linear ways on the environment state. However, once the setter figures out the task structure, it is more easily able to train the solver, and so it surpasses desired distribution training to reach asymptotic mastery sooner. Furthermore, the fact that the desired tasks are somewhat feasible early in learning means that the targeted setter has less of an advantage over the regular setter.

### 4.4 COMPARISON TO PRIOR WORK

### 4.4 Comparison to prior work

We compare to the Goal GAN (Florensa et al., 2017), which is the closest to our approach. Our notion of goal feasibility is related to their binary partitioning of goals into those that are of intermediate difficulty, and those that are not. However, our continuous notion of feasibility has advantages: it allows uniformly sampling feasibility, can be estimated from one run per goal, and may be easier to learn. Furthermore, while their discriminator may implicitly encourage increasing coverage and identifying valid goals, training GANs can be difficult, and our explicit losses may be more effective.

We implemented an asynchronous version of the algorithm outlined in Florensa et al. (2017), which continuously trains the GAN and the agent, rather than iterating between training each. This allowed us to equalize computational resources between approaches, and apply their approach to the same distributed solver agent we used, in order to have as fair a comparison as possible. See appendix D for details. We first demonstrate that our implementation of their approach achieves similar performance to our method on a simple $(x, y)$ location-finding task like that used in their paper, but implemented in our more complex 3D environment (fig. 6a), and learnt from pixels rather than state. However, we show that on our more complex color-finding tasks their approach is not as successful as ours (fig. 6b-c, and supp. fig. 8). Furthermore, maintaining and sampling from a large memory buffer, and running the agents on each goal multiple times to get a label of whether it was intermediate difficulty were very costly, and their approach thus required more memory and wall-clock time than ours for an equivalent amount of agent steps. In addition, the instabilities introduced by the adversarial training resulted in less consistent results from their approach even on the simple location finding task.

Overall, our results suggest that our approach is more stable and is more effective on complex tasks. Furthermore, as noted above, Florensa et al. did not attempt the challenge of curriculum generation in environments that vary (which is why we did not compare to their algorithm in the alchemy environment), while we have also demonstrated success in that setting.

## 5 DISCUSSION

In this paper we outlined a strategy for automated curriculum generation for goal-conditioned RL agents in complex environments. The curriculum is generated by training a setter to propose goals for a solver agent. The setter is trained to choose goals based on their validity, feasibility and coverage, and we demonstrated that all three of these components are necessary in a complex environment. Furthermore, we showed that this approach substantially outperforms a prior approach and baselines on complex tasks, including 3D environments with rich visual experiences, interactions with objects, and complex control (a nearly 50-dimensional action space). These results represent a substantial step towards automated curriculum generation in rich environments.

We also highlighted the necessity of employing curriculum generation in environments that vary from episode to episode. To address this challenge, we demonstrated that by providing an environmental observation to the setter and judge, our algorithm can learn to generate reasonable curricula in variable environments. This approach outperformed a lesioned version without the environmental observation and other baselines, and nearly reached the performance of an oracle curriculum based on the true task distribution (where available). To our knowledge, these are the first results to demonstrate automated curriculum generation for goal-conditioned agents in environments where the possible goals vary from episode to episode. This is an important step towards developing automated curricula in environments with complexity closer to the real world.

However, our work also highlights challenges when the environment varies. Learning a conditional generative model for the setter in combinatorially complex environments like our alchemy setting can be challenging. From only a generative loss, it is difficult for the model to learn how to extract the appropriate latent variables from an observation and manipulate them appropriately. Training setters in rich environments may require auxiliary information about the structure of the world, or breakthroughs in conditional generative modelling. This is an important direction for future work.

Finally, we pointed out the challenge of efficiently achieving competence on desired goals which are distributed in a small region of goal space. We demonstrated a loss that can help to solve this problem by targeting the setter toward the desired goal distribution.

Overall, we showed the success of our setter-solver approach in rich environments, and extensions that allowed it to work in complex tasks with varying environments and guide the solver efficiently towards mastering desired goals. Although work remains to be done, we believe that the strategies we have outlined here will be a useful starting point for automatically devising curricula for RL agents in the increasingly complex tasks we desire them to solve.

## 6 ACKNOWLEDGEMENTS

We would like to thank the DeepMind Worlds Team for their help with the environments used in this paper.

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

# A  SUPPLEMENTAL FIGURES

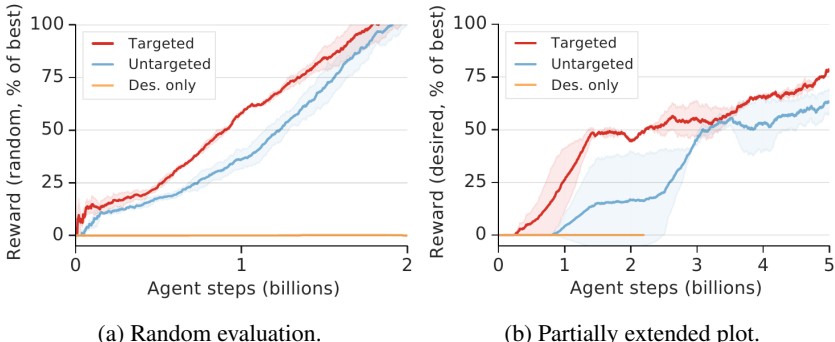

(a) Random evaluation.

(b) Partially extended plot.

Figure 7: Supplemental plots for figure 5b. (a) Random evaluation is not impaired by the desirability loss, suggesting that in this setting it is actually helping the setter to expand goals in a useful direction. (b) Partial continuation of figures 5b, showing the advantages persist. However, some of the experiment versions were killed by a server issue around 2 billion steps, which is why the figure in the main text terminates there. This panel plots runs with slightly worse hyperparameter settings (weight $\beta_{des.}$ of 1 instead of 5), but that ran for longer.

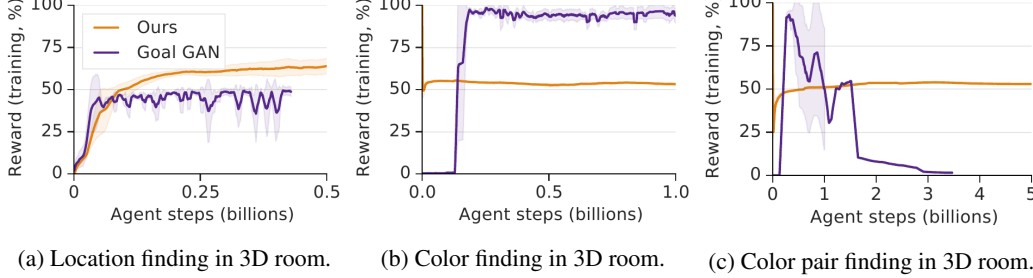

(a) Location finding in 3D room.

(b) Color finding in 3D room.

(c) Color pair finding in 3D room.

Figure 8: Comparing scores agents obtain on the training levels proposed by our method and Goal GAN. In the location task (a) both methods generate levels of intermediate feasibility/difficulty. However, on more complex tasks (b-c), while our method continues to generate tasks of intermediate feasibility (on average), the goal GAN tends to overshoot and choose tasks that are too easy. This may be because the capacity of the discriminator is mostly used in determining which tasks are valid (since most tasks are impossible in these more complex environments), and so it struggles to find valid but difficult tasks. By contrast, our judge is mainly responsible for determining feasibility, while the validity should mostly be handled by the setter losses. This frees up the capacity of the judge for accurately estimating feasibility. Furthermore, the fact that we are proposing levels at varying levels of feasibility will like give the judge more accurate data to train on. Finally, the adversarial training of Goal GAN results in noisier performance even in the simpler task.

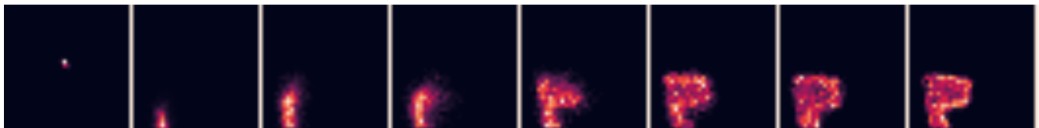

Figure 9: Visualizing the generated curriculum for the location-finding task which we used in section 4.4. We chose to visualize the curriculum for this task because it is in a lower-dimensional and easier-to-understand task space than the main tasks we considered. In this plot we show a heat map for the goal locations selected by the setter over the course of learning. The agent is placed in an L-shaped room. Goals are chosen by specifying two coordinates where the agent needs to go. We assume we have a rough idea of a bounding box for the coordinates of reachable locations in the room, and use those bounds to limit the goals sampled by the setter. The L-shaped room is actually placed in the bottom left part of this bounding box, and the setter needs to discover this. The first picture above shows how at random initialisation, the setter chooses locations which are not inside the room. Our validity loss will then help move these locations inside the room, as shown in the second picture. After this, the setter gradually expands the set of sampled locations as the agent learns, until they fill the whole room, as shown in the progression to the picture on the right.

## B  ARCHITECTURE & TRAINING DETAILS

**Solver**  The solver consists of a ResNet (3 sections, each with 2 residual blocks, with 16, 32, and 32 filters respectively, all of size 3 and stride 1) for vision and a goal input (processed through a linear transformation) to a core LSTM (256 hidden units). The output of the LSTM is fed into separate MLPs for policy (1 hidden layer, 64 hidden units) and value (1 hidden layer, 128 hidden units) outputs. The judge is an MLP (3 hidden layers, 64 units each).

**Setter**  The setter consists of a Real-Valued Non-Volume Preserving (RNVP) network (Dinh et al., 2016), which has the useful property of providing access to exact log-likelihood of a sample, i.e. exact invertibility. The basis of this invertibility is a clever trick used in the RNVP design, where at each layer only half of the latent variables are updated, but their update depends multiplicatively on the other half of the latent variables. This allows for representing complex transformations, but because the conditioning latent variables are preserved in the output of the layer, it can be exactly inverted. By alternating between updating different subsets of the latent variables at each layer, the variables can undergo a complex evolution, with that evolution still remaining precisely invertible. For more details see sec. 3.3 and fig. 2 of Dinh et al. (2016).

The setter has 3 blocks. Each block is a fully connected Highway Network (Srivastava et al., 2015) with 3 layers, and a hidden size of 32, 128, or 512, depending on the complexity of the generative task. Specifically, for the basic color-finding tasks (both single colors and pairs), the hidden size was 32, for the recolored color-finding tasks it was 128, and for the alchemy tasks it was 512. Non-linearities in all networks are leaky ReLUs, except when generating goals, where arctans were used to get outputs between -1 and 1, which were then scaled appropriately for the goal domain.

**Desirability discriminant**  As in Arjovsky et al. (2017), we ensure that $D$ is Lipschitz by clipping the parameters of the discriminator $D$ to the range $[-0.1, 0.1]$ before each setter training step.

### B.1  OBSERVATION-CONDITIONED SETTER & JUDGE

When the setter and judge are observation-conditioned, their vision architecture is the same as the agent ResNet, and the setter and judge share its weights (but do not share weights with the agent vision architecture). In the setter, the conditioning information, including the desired feasibility $f$ and the output of the ResNet (where applicable) is concatenated with the part of the latent being transformed in each layer. In the judge, the output of the ResNet, if applicable, is concatenated with the goal before the goal is input to the judge. We found it was useful to down-weight the vision information by fixed constants before inputting it to the setter and the judge, likely because it allowed them to learn first to respond to the goals alone before trying to incorporate visual input. These constants were determined via a hyperparameter sweep, and were 0.1 for the setter in all

conditioned tasks, and $10^{-7}$ and $10^{-6}$ respectively for the judge in the alchemy tasks and recolored color-finding tasks. (Despite the small weight on the visual input, the judge appears to still use it, as the final solver performance is worse if the weight is reduced by another order of magnitude, or increased.)

## B.2 TRAINING

The solver agents were trained using the framework of Espeholt et al. (2018), with the RMSProp optimizer, without momentum and a learning rate of $2 \cdot 10^{-4}$. The setters were trained using Adam, with learning rates of $2 \cdot 10^{-4}$ on the 3D tasks and $3 \cdot 10^{-4}$ on the grid-world alchemy tasks.

The training setup is described in fig. 1 at a high level. We provide now more details. We use a distributed setup that is identical to Espeholt et al. (2018), except for the addition of a learner to train the Judge and Setter. We therefore end up with three types of workers, that run asynchronously and communicate data to each other. Below, we write in pseudo code what loops are running on each type of worker. We have written those for the conditional setter case.

---

**Algorithm 1:** Solver-Actor loop

// The Solver-Actor collects data and send them to the learners;
Start environment;
Sample $feasibility$ uniformly in $[0, 1]$;
Get $first\_observation$ from environment and sample $goal$ from Setter with given $feasibility$;
**while** *True* **do**
    $trajectory = []$;
    **for** *n = 1...N* **do**
        Get $observation$ from environment;
        Choose $action$ using the agent and $observation$;
        Apply $action$ to environment;
        Get $reward$ and $discount$;
        $trajectory$ += $(observation, reward, discount, action)$;
        **if** *this was the last step of the episode* **then**
            Send $(first\_observation, goal, reward)$ to the Setter-Learner;
            Restart environment to start new episode;
            Sample $feasibility$ uniformly in $[0, 1]$;
            Get $first\_observation$ from environment and sample $goal$ from Setter with given $feasibility$;
    **end**
    Send trajectory to the Solver-Learner;
    Request and apply updated weights for agent from Solver-Learner;
    Request and apply updated weights for setter from Setter-Learner;
**end**

---

**Algorithm 2:** Setter-Learner loop

// The Setter-Learner receives data from the Solver-Actor and use it to train the Judge and Setter;
Initialise weights of Setter and Judge;
**while** *True* **do**
    Wait until we have received a batch of tuple $(first\_observation, goal, reward)$;
    Use batch to train Judge with a gradient update;
    Do a gradient update on setter using batch of $first\_observation$ as conditioning;
**end**

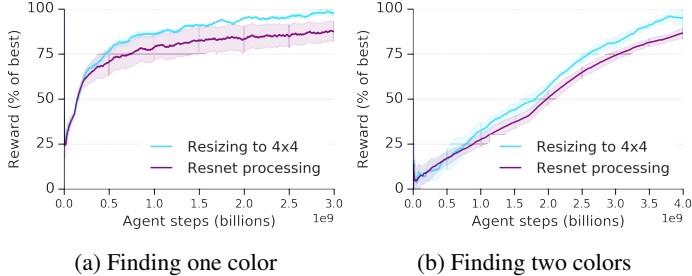

(a) Finding one color        (b) Finding two colors

Figure 10: Learning a good latent representation from a conditioning observation can be difficult. For the recolored find colour task, the setter only needs to extract the two dominant colours in the room from the observation in order to choose valid goals. We can use this knowledge to hard-code a better representation for the conditioning observation by simply resizing the observation to a $4 \times 4$ image, instead of processing it with a deep ResNet. This representation will be closer to the true latent variables that determine the valid goals in the episode. When we do this, the agent learns more rapidly, suggesting that this is indeed a better representation for conditioning the setter (and judge). However, choosing this representation requires using prior knowledge about the task structure. This knowledge might not always be available.

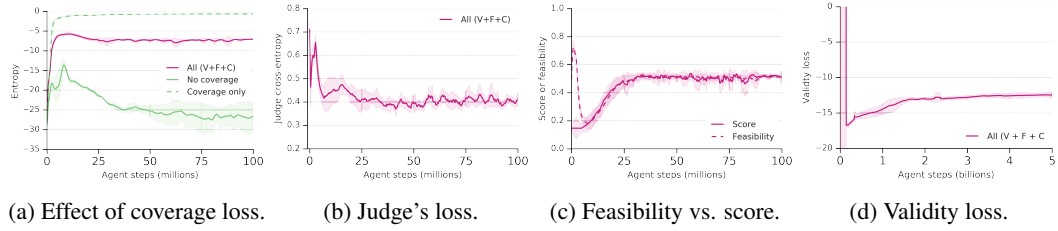

(a) Effect of coverage loss.    (b) Judge's loss.    (c) Feasibility vs. score.    (d) Validity loss.

Figure 11: Examining the indivudal losses. All curves are for the Color pair finding task. Note how curves (a) to (c) are only up to 100 million steps. This is still early in training, but the losses are very stable after that point, as the setter and judge continually adapt to the solver's improvements. Curve (d) shows data for the whole training as the loss slowly increases throughout. Purple curves are averaged over nine runs. Green curves are averaged over three runs. All curves show standard deviation.

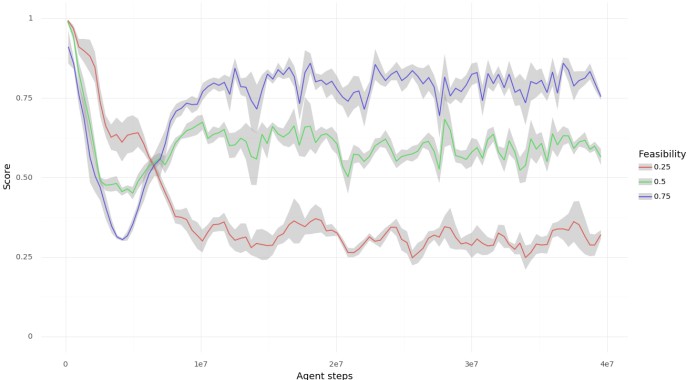

Figure 12: The setter is learning to appropriately condition on the feasibility input to produce tasks of varying difficulty. This plot shows the average score (where the maximum score is 1) of the solver on 1-color finding tasks generated by a setter for feasibility inputs of 0.25, 0.5 and 0.75. Each graph is averaged over 3 setter/solver pairs. We only show the behaviour early on in training, as it is stable throughout. The setter is producing tasks close to the desired input feasibility, so that uniformly sampling feasibility causes it to produce tasks of varying difficulty.

---

**Algorithm 3:** Solver-Learner loop

---

// The Solver-Learner is identical, and uses the same losses, as a learner from Espeholt et al. (2018);
Initialise weights of Solver (agent);
**while** *True* **do**
    | Wait until we have received a batch of trajectories from Solver-Actor;
    | Do a gradient update using the batch;
**end**

---

### B.3 EXAMINING THE INDIVIDUAL LOSSES

Our experiments demonstrate that our setup leads to better agent performance. In this section, we examine the components of our system, to evaluate whether they are performing as intended.

First, we look at coverage, which we measure using entropy. In the color pair finding task, the setter defines a distribution in the space $[0, 1]^6$. The uniform distribution is a maximum entropy distribution on this space, with an entropy of $0$. Entropy is a particularly easy loss to maximise. Indeed, looking at Fig. 11a, we see it immediately climbs to a maximal value, before slowly decreasing as the pressure from the two other losses builds up. In the same figure, we see that removing the coverage loss leads to lower entropy early on, and a larger collapse later. This means the setter will offer a far less varied set of tasks for the solver to train on without the coverage loss. Finally, we observe how using only the coverage loss leads to near maximum entropy being reached very quickly (the entropy then keeps on slowly climbing during training). However, by ignoring validity and feasibility, most the tasks the setter proposes are impossible for an expert, or at least impossible for the agent at present. Thus the overall performance is worse.

While the feasibility loss is being optimised as soon as training start, this loss can only become useful once the judge has learnt how to measure feasibility efficiently. Despite training on a non-stationary distribution, we expect the supervised training of the judge to result in quicker convergence. We can indeed see in Fig. 11b that after some very rapid oscillations, the judge's cross-entropy loss drops very quickly. Another indication that the judge is indeed training well, is that if we compare the average score of the agent on tasks generated by the setter to the average feasibility of those same tasks as measured by the judge, we see in Fig. 11c these two values match very quickly. Finally, we note that the setter is learning to produce tasks of approximately appropriate feasibility, as shown in fig. 12. The solver is performing slightly better than expected, presumably because of the lag in training the judge and solver, but the difference is not too great, suggesting that the lag is not too bad.

Finally, we see the validity loss in Fig. 11d drops very quickly to a minimum value when training starts, and slowly increases throughout training. We see this as a good behavior since the hindsight provided by this loss is most useful early on when the setter needs to choose which are reasonable tasks to train on even if it has not yet learnt much about the structure of tasks.

## C ENVIRONMENT & TASK DETAILS

### C.1 COLOR FINDING

The room is L-shaped, which means that some objects may be hidden from some locations in the room. The objects present consist of pillow-like objects, a bookshelf, chair, a few wall-mounted shelves and table. The furniture is placed randomly along the walls of the room at the beginning of each episode. The pillows are randomly placed throughout the room, with some scattered on the floor, and some placed on shelves.

The actions for this environment consist of multiple speeds of moving forward, back, left, and right, rotating left and right, looking up and down, grabbing objects and moving with them, and manipulating held objects, rotating them along various axes or moving them closer or farther away. This results in a total of 45 actions. The agent has to choose 15 actions per second. Observations are in the form of $72 \times 96$ RGB pixel images. Each pixel value was normalised in the range $[0, 1]$.

For single color finding, the agent received a reward of 1 if the color averaged over an $8 \times 8$ patch in the center of the screen was within an $\ell_2$ distance of $\epsilon = 0.1$ of the goal color $\in [0, 1]^3$. For color pair finding, the agent received a reward of 1 if the color in an $8 \times 8$ patch left of the center of the screen was within $\epsilon$ of the first goal color, and similar with a patch right of the center of the screen and the second goal color. That is, the agent needed to get both colors correct to receive any reward on the pair color finding task. If the agent received a non-zero reward the episode would terminate, otherwise the episode would terminate with a reward of 0 after 500 environment steps.

**Desired distribution:** Our desired distribution for the targeting experiments consisted of pairs of 12 colors: the 3 primary colors and 3 secondary colors, and slightly more muted shades of these (all components moved by 0.3 towards the middle). We found $\beta_{des.} = 5$ to be optimal, though results in fig. 7b are from runs with $\beta_{des.} = 1$.

### C.2 GRID-WORLD ALCHEMY

The actions consist of movement in the four cardinal directions. The room is a $9 \times 9$ grid surrounded by an impassable wall of size 1 (for a total grid size of $11 \times 11$), with four objects randomly placed in it. Each object has two colors which are randomly sampled uniformly from $[0, 1]^2]$ — only the red and blue components were sampled, with the green component fixed at zero. This makes the conditional generative problem slightly easier. The agent receives visual input in which every grid square is rendered as $2 \times 2$ in order to have the two-colored objects, i.e. it received visual input of size $22 \times 22$.

To avoid trivial solutions for the setter, it was necessary to avoid rewarding the solver (or training the setter's possibility loss) if the agent failed to pick up an object.

**Desired distribution:** Our desired distribution for the targeting experiments consisted of the most difficult tasks in this world: combining half of the objects in the level. We found $\beta_{des.} = 1.5$ to be optimal for this task.

## D DETAILS OF COMPARISON TO FLORENSA ET AL. (2017)

In order to compare to the Goal GAN approach proposed by Florensa et al. (2017), it was necessary to make a few changes. We wanted to make as fair a comparison as possible, so we wanted to use their approach to train the same distributed solver agent we used. In order to do this, we had to modify their algorithm to run asynchronously. Specifically, instead of alternating between training the GAN and training the agent, we trained both simultaneously in separate processes. Because of the asynchronous approach it was also difficult to have a single unified memory buffer; instead each copy of the agent had its own memory buffer which could hold up to 10,000 goals, and added goals to it with a probability of 0.01 rather than performing an expensive comparison to the prior goals at each step. As in the original paper, we sampled $1/3$ of the goals from the memory buffer, and $2/3$ from the setter. Even with our modifications and the simpler MLP architecture (see below), their approach required more computation time than ours for the same number of agent steps.

As in our RNVP architecture, we use a latent noise vector sampled from a standard normal distribution of the same dimensionality as the goal outputs. We originally tried implementing their GAN with the same RNVP architecture we used for our setter (see above), but we had substantial issues with mode collapse, so we switched to using an MLP architecture like was used in their original paper. We used 3 hidden layers with 64 units each in the location tasks, and 128 units each in the color tasks, and for the discriminator we used 3 hidden layers with 128 units in both tasks. The GAN was trained via the Adam optimizer with a learning rate of $5 \cdot 10^{-4}$. All these hyperparameter were determined by a sweep.

## E MISCELLANEOUS

Color palettes for plots were modified from Harrower & Brewer (2003).

