# OpenReview forum: "Automated curriculum generation through setter-solver interactions"
_ICLR.cc/2020/Conference — Accept (Poster)_

### Official Review · AnonReviewer1 · 2019-10-22
**Official Blind Review #1**

**Rating:** 6

**Review:**

The paper studied the problem of reinforcement learning under the sparse or dynamic rewarding environment.
The authors propose a promising automated curricula generation scheme, which considers goal validity, goal
feasibility, and goal coverage to construct useful curricula for the underlying agents. Rewards and loss functions
are proposed individually for the solver, judge, and setter. Empirical studies demonstrate the capability of the
proposed model in generating task curricula across several complex goals. In general, I believe the studied
problem is interesting, and the proposed model is promising.  However, I am not familiar with the curricula
generation in the reinforcement learning setting. All I can say is the approach is intuitively appealing, the text is
well written and easy to follow, even for an outsider. A minor concern is about the experiments. Most of the
experiments are presented in the metric of Reward (% of best). It would be helpful if the authors can conduct
some illustrative case studies (better in different scenarios) to show some specific tasks, the generated task-oriented
curriculum, and provide some intuitive discussion to show the readers why the generated curriculum is beneficial
for the agent.

**Experience Assessment:**

I have read many papers in this area.

**Review Assessment: Checking Correctness Of Derivations And Theory:**

I carefully checked the derivations and theory.

**Review Assessment: Checking Correctness Of Experiments:**

I carefully checked the experiments.

**Review Assessment: Thoroughness In Paper Reading:**

I read the paper at least twice and used my best judgement in assessing the paper.

---

> ### Author Response · Authors · 2019-11-07
> **Response to your review**
>
> Thank you for the thoughtful review! We appreciate your comments. We’ve included a plot of the progression of the curriculum on the 2D location finding task in the appendix, which shows how the setter first learns to propose locations closest to the agent, and then gradually expand them throughout the entire valid space the agent can explore. We hope that this is a useful case study for seeing that our setter algorithm is behaving basically sensibly.
>
> It is more difficult to visualize the curricula for the color tasks, since the task spaces are higher dimensional. However, we think that there are intuitive reasons a curriculum might work in the color-pair finding tasks. The setter could learn to first generate matching colors in the two pairs at first, to effectively reduce the problem to the easier one-color finding problem initially. It could then first generate simple goals for the agent to achieve that are relatively easy, such as looking at the ceiling or the floor. Then it could move on to generate more complex goals, such as finding different colors that only appear on rare objects within the room, and to finding simple pairs, such as a two colors that always appear next to each other, like a piece of furniture next to a wall. This would finally allow for learning the most complex combinations, one rare color next to another, one of which is demonstrated in the video linked from the paper. Hopefully this helps guide your intuitions.

---

### Official Review · AnonReviewer2 · 2019-10-23
**Official Blind Review #2**

**Rating:** 6

**Review:**

This paper proposes an autocurricula scheme to train a goal-conditional agent in a dynamic and sparse-rewarding environment. The main idea is to train a setter model to sample goals for next-step training, where the setter can make the decision either based on the training history or the environmental observation (conditional case). The paper proposes three criteria which leads to three types of loss to train the setter model, i.e., goal validity (the goal should be achievable by some existing policy), goal feasibility (how probable the current policy can achieve the goal), and goal coverage (the sampled goals by the setter need to cover all possible goals). A judge model is needed to output the feasibility of a given goal. So the autocurricula scheme contains the solver (agent), the setter, and the judge, each having its own combination of loss and they are trained together. Given a desired goal distribution, the paper proposes to additionally train a discriminator whose optimization objective is Wasserstein loss. In experiments, they evaluate the proposed method on three types of tasks in two environments, i.e., 3D color finding and grid world alchemy. The goals in the two environments are similar in that they all aim to achieve some color or color pairs. The difference lies in that the first one finds colors while the second pick up colors. Each environment can be changed between episodes by changing the colors of objects in the scenes. Experimental results show that different combinations of the three types of losses can bring improvements in some scenarios. Making setter and judge conditioned on environment observation can further improve the success rate. Given a desired distribution of goals, the learning becomes more efficient. The paper compares this method with Goal GAN as a baseline and outperforms it on the three tasks.

This paper is an early exploration of the effectiveness of autocurricula on goal-conditional tasks. The experiments to some extent verify the effectiveness of the proposed method. The high-level idea of setter-solver learning can be possibly generalized to other tasks. However, it is not convincing that the detailed techniques proposed in this paper can be easily generalized to more complicated environments and tasks. The writing of this paper is not clear and enough and the whole paper is difficult to understand. The experimental comparison is insufficient since only one baseline of curriculum (goal GAN) is compared (some of the methods mentioned in Section 2 can be easily applied though some of them were not specifically designed for goal-conditional tasks).

Detailed comments:

1) There are too many details and hyperparameters involved when applied to specific tasks as shown in this paper. Considering the two environments in this paper is easier than most game environments studied in other papers (many of them also uses some types of the curriculum), the experimental result is not very strong. The proposed scheme is possible to be too simple for a slightly more complicated environment or task.

2) The success of training relies on interactions between the three to four types of losses, but they all have the same weights in the combined loss. Is there any special reason for not setting different weights for different losses?

3) It is not clear how the inverse transform of the setter model (i.e., S^{-1} in L_val) is achieved.

4) There are three random quantities involved in the loss terms, i.e., the noise \xi applied to the sampled goals, the latent input z, the feasibility f. It is not clear whether the randomness on them will dominate the curriculum or not.

5) The setter model takes z and f as inputs, where z is a vector but f is a scalar. Will it result in a setter model whose output does not change too much when changing f?

6) How does the proposed method compare to simple goal selection by uncertainty, hardness, or curiosity? How does it compare to hindsight experience reply methods?

7) It is helpful to visualize the generated curriculum, i.e., the trajectory of selected goals during training in the 2D/3D grid.

8) Computing L_judge requires to test whether the agent can finally achieve the sampled goals. Is it too computational expensive during training?

---------------------------

Update after rebuttal:

Thanks for the detailed reply from the authors! They answer most of my questions. I think this paper has a moderate contribution and thus will keep my "weakly accept" rating.

**Experience Assessment:**

I have published one or two papers in this area.

**Review Assessment: Checking Correctness Of Derivations And Theory:**

I carefully checked the derivations and theory.

**Review Assessment: Checking Correctness Of Experiments:**

I carefully checked the experiments.

**Review Assessment: Thoroughness In Paper Reading:**

I read the paper thoroughly.

---

> ### Author Response · Authors · 2019-11-07
> **Response to your review (Part 1)**
>
>
> Thank you for the detailed and thoughtful review. We respond to your comments inline below.
>
> 1) There are too many details and hyperparameters involved when applied to specific tasks as shown in this paper. Considering the two environments in this paper is easier than most game environments studied in other papers (many of them also uses some types of curriculum), the experimental result is not very strong. The proposed scheme is possible to be too simple for a slightly more complicated environment or task.
>
> We believe our results represent a substantial increase in complexity beyond the environments used in prior work. Indeed, our comparison to a prior paper in section 4.4 (and the analysis of it in supplemental fig. 8) shows that in simple tasks like those used in the prior work their algorithm performs comparably to ours, but that the complicated version of our tasks are complex enough that their algorithm fails. Recent exploration-based pretraining, like Diversity is All You Need, has also used relatively simple environments, such as simple control tasks. Furthermore, we are not aware of any prior work that considers goal-conditioned tasks with variable environments, or targets desired goals. While we have not demonstrated these results on a task as complex as starcraft, say, we believe they nevertheless represent a substantial increase in the complexity of the environments on which automated curriculum generation schemes have been demonstrated. We hope that our paper, if accepted, will help to inspire further work tackling even more complex tasks.
>
> 2) The success of training relies on interactions between the three to four types of losses, but they all have the same weights in the combined loss. Is there any special reason for not setting different weights for different losses?
>
> We found that this was not necessary for the three basic losses, and so we did not add these additional hyperparameters. However, we did need to increase the weight slightly for the desirability loss in order for it to have a similar magnitude to the other losses and perform optimally. This is the hyperparameter Beta_{des} noted in the description of the desired goal distributions loss, and the exact values used are given in appendix C.
>
> 3) It is not clear how the inverse transform of the setter model (i.e., S^{-1} in L_val) is achieved.
>
> The setter model is an Real-valued Non-volume preserving generative model (https://arxiv.org/pdf/1605.08803.pdf), one of the benefits of this model class is that it allows exact inference of the latents given a sample (i.e. exact inversion of the model). We have added a section in the appendix briefly explaining how and why these models are invertible.
>
> 4) There are three random quantities involved in the loss terms, i.e., the noise \xi applied to the sampled goals, the latent input z, the feasibility f. It is not clear whether the randomness on them will dominate the curriculum or not.
>
> The noise applied to the sampled goals is quite small, and is only used for training the setter. While the randomness of z and f presumably contributes to the success of the curriculum, we would like to note a few observations that suggest it is not the primary driving factor: 1) Uniformly randomly sampled tasks result in little to no solver learning in our environments (this is shown in Fig. 4 for the observation conditioned case, e.g., and we also observed it in the basic tasks). 2) The coverage loss alone, which increases the randomness of the setter outputs, does not alone result in any learning on the color pair finding or alchemy tasks (although it is reasonably successful in the single-color-finding task). 3) Removing the coverage loss results in no learning in the color-pair finding environment, suggesting that the randomness of the setter inputs is not enough to result in good learning without equally random outputs. 4) As we show in our response to your next question, f is not just a source of randomness, but the setter is actually using it to choose tasks of variable difficulty for the solver.
>
> (Continued in next comment, due to character limit.)

---

> > ### Author Response · Authors · 2019-11-07
> > **Response to your review (part 2)**
> >
> > 5) The setter model takes z and f as inputs, where z is a vector but f is a scalar. Will it result in a setter model whose output does not change too much when changing f?
> >
> > Although it is only a single scalar, the feasibility is used in the definition of a loss for the setter, which provides a strong learning signal. We include a plot below that demonstrates this learning (and have added it to appendix B, in the section where we examine the individual losses). Unfortunately we can’t include the figure in this comment, but please see it in the paper. The plot shows the average score (where maximum score is 1) of the solver on 1-color tasks generated by a setter for feasibility inputs of 0.25, 0.5 and 0.75. Each graph is averaged over 3 setter/solver pairs. We only show the behaviour early on in training, as it is stable throughout. The plot clearly shows that the setter quickly learns how to condition appropriately on the feasibility. The solver is slightly outperforming the nominal values, but this is to be expected as it is continually learning.
> >
> >
> > 6) How does the proposed method compare to simple goal selection by uncertainty, hardness, or curiosity? How does it compare to hindsight experience reply methods?
> >
> > We compare to various lesions of our method, including all non-empty subsets of validity, feasibility, and coverage in section 4.1/fig. 3. We also compare to a prior approach to choosing moderately hard goals in section 4.4. Hindsight experience replay would be a useful additional comparison, however, the setter validity loss is effectively trying to give the setter hindsight, in the hopes that the agent will learn from it.
> >
> >
> > 7) It is helpful to visualize the generated curriculum, i.e., the trajectory of selected goals during training in the 2D/3D grid.
> >
> > We have generated a visualization of the curriculum on the location-finding task, and added it to the supplemental material (appendix A). It is more difficult to visualize the curricula for the color tasks, since they are in higher-dimensional and less-intuitive spaces. The figure shows a heat map for the goal locations selected by the setter through learning. The agent is placed in an L-shaped room. Goals are chosen by specifying two coordinates where the agent needs to go. We assume we have a rough idea of a bounding box for the coordinates of reachable locations in the room, and use those bounds to limit the goals sampled by the setter. The image at the right below shows that the L-shaped room is actually placed in the bottom left part of this bounding box, and the setter needs to discover this. The first picture below shows how at random initialisation, the setter chooses locations which are not inside the room. Our validity loss will then help move these locations inside the room, as shown in the second picture below. After this, the setter gradually expands the set of sampled locations, until they fill the whole room, as shown in the picture on the right.
> >
> >
> > 8) Computing L_judge requires to test whether the agent can finally achieve the sampled goals. Is it too computational expensive during training?
> >
> > The judge is trained based only on the trajectories generated by the solver during learning, and whether or not the agent achieved the given goal in that trajectory. It’s necessary for the agent to experience the environment during RL training, so the judge does not require collecting any additional data for the solver beyond what the solver already requires. Since the judge is a relatively small network, training it with the data that has already been collected for the solver is quite computationally cheap.

---

### Official Review · AnonReviewer3 · 2019-10-24
**Official Blind Review #3**

**Rating:** 6

**Review:**

This paper tackles the task of automatically inducing a curriculum for agents learning through reinforcement. Specifically, they use two agents — a setter agent that sets goals, and a solver agent that solves the goals provided by the setter.  While this has been explored before, the difficulty lies in training both agents simultaneously in a robust fashion. If the goals are too difficult, the solver will be unable to solve them and if they are too easy, the solver will be unable to improve. The authors propose a combination of different losses to help the setter balance its goal predictions — validity, feasibility and coverage. In addition, they train a judge model predict the reward that the solver agent would achieve on a goal proposed by the setter. Empirical results on two setups demonstrate the effectiveness of this approach in learning a good curriculum.

Pros:
1. Clear writing, method is easy to understand.
2. Novel objectives for a multi-agent training setup

Cons:
1. Empirical results do not contain any baselines or prior work comparisons (only ablations of the proposed model)

——
Updates:
Thanks to the authors for their response. I realize I was not very clear in my comment above. I wanted to point out that the authors could consider adding other (simpler) baselines such as Sukhbaatar et al. (2017) to make their empirical results more complete for the navigation tasks (even though these methods make certain assumptions, it would be interesting to see how much of an effect they have compared to the proposed method). If you are able to, I’m convinced the paper will be much stronger. Nevertheless, I think this is very interesting work!

**Experience Assessment:**

I have read many papers in this area.

**Review Assessment: Checking Correctness Of Derivations And Theory:**

N/A

**Review Assessment: Checking Correctness Of Experiments:**

I assessed the sensibility of the experiments.

**Review Assessment: Thoroughness In Paper Reading:**

I read the paper at least twice and used my best judgement in assessing the paper.

---

> ### Author Response · Authors · 2019-11-07
> **Response to your review**
>
> Thank you for the thoughtful comments. We would like to highlight that we empirically compare our basic approach to prior work in section 4.4, and show that in complex environments it outperforms a recent approach to automated curriculum generation for goal-conditioned RL. In fig. 8 in the supplemental material, we explore part of the reason for this, showing that our approach to curriculum generation more reliably produces tasks that challenge the solver in the complex environments.
>
> We are not aware of any other prior work that can generate automated curricula for goal-conditioned RL in a variable environment, or that optimizes the curriculum towards a desired goal distribution, and so we were unable to compare to prior work in that setting. We think part of the contribution of our paper is to highlight these challenges and hopefully inspire further work in this area.

---

### Decision · Program_Chairs · 2019-12-19

**Decision:**

Accept (Poster)

**Comment:**

The authors introduce a method to automatically generate a learning curriculum (of goals) in a sparse reward RL setting, examining several criteria for goal setting to induce a useful curriculum.  The reviewers agreed that this was an exciting research direction but also had concerns about baseline comparisons, clarity of some technical points, hyperparameter tuning (and the effect on the strength of empirical results), and computational tractability.  After discussion, the reviewers felt most of these points were sufficiently addressed.  Thus, I recommend acceptance at this time.